# Red-Emitting Latex Nanoparticles by Stepwise Entrapment of β-Diketonate Europium Complexes

**DOI:** 10.3390/ijms232415954

**Published:** 2022-12-15

**Authors:** Hwan-Woo Park, Daewon Han, Jong-Pil Ahn, Se-hoon Kim, Yoon-Joong Kang, Young Gil Jeong, Do Kyung Kim

**Affiliations:** 1Department of Cell Biology, Konyang University College of Medicine, Daejeon 35365, Republic of Korea; 2Department of Business Cooperation Center, Korea Institute of Ceramic Engineering and Technology, Bucheon 14502, Republic of Korea; 3Biomaterials & Processing Center, Korea Institute of Ceramic Engineering and Technology, Cheong-ju 28220, Republic of Korea; 4Department of Biomedical Science, Jungwon University, Geosan 28023, Republic of Korea; 5Department of Anatomy, College of Medicine, Konyang University Hospital, Daejeon 35365, Republic of Korea

**Keywords:** polystyrene, europium complex, β-diketone, photoluminescence, LFIA

## Abstract

The core–shell structure of poly(St-*co*-MAA) nanoparticles containing β-diketonate Eu^3+^ complexes were synthesized by a step-wise process. The β-diketonate Eu^3+^ complexes of Eu (TFTB)_2_(MAA)P(Oct)_3_ [europium (III); 4,4,4-Trifluoro-1-(2-thienyl)-1,3-butanedione = TFTB; trioctylphosphine = (P(Oct)_3_); methacrylic acid = MAA] were incorporated to poly(St-*co*-MAA). The poly(St-*co*-MAA) has highly monodispersed with a size of 300 nm, and surface charges of the poly(St-co-MAA) are near to neutral. The narrow particle size distribution was due to the constant ionic strength of the polymerization medium. The activated carboxylic acid of poly(St-*co*-MAA) further chelated with europium complex and polymerize between acrylic groups of poly(St-*co*-MAA) and Eu(TFTB)_2_(MAA)P(Oct)_3_. The *E_m_* spectra of europium complexes consist of multiple bands of *E_m_* at 585, 597, 612 and 650 nm, which are assigned to ^5^D_0_→^7^F_J (J = 0–3)_ transitions of Eu^3+^, respectively. The maximum *E_m_* peak is at 621 nm, which indicates a strong red *E_m_* characteristic associated with the electric dipole ^5^D_0_→^7^F_2_ transition of Eu^3+^ complexes. The cell-specific fluorescence of Eu(TFTB)_2_(MAA)P(Oct)_3_@poly(St-*co*-MAA) indicated endocytosis of Eu(TFTB)_2_(MAA)P(Oct)_3_@poly(St-*co*-MAA). There are fewer early apoptotic, late apoptotic and necrotic cells in each sample compared with live cells, regardless of the culture period. Eu(TFTB)_2_(MAA)P(Oct)_3_@poly(St-*co*-MAA) synthesized in this work can be excited in the full UV range with a maximum *E_m_* at 619 nm. Moreover, these particles can substitute red luminescent organic dyes for intracellular trafficking and cellular imaging agents.

## 1. Introduction

The most representative POC (point-of-care) testing technology is LFIA (lateral flow immunoassay), which is widely used in various fields such as rapid diagnosis, environmental monitoring, veterinary, pharmaceutical analysis, etc. [1]. In particular, epidemics occurring worldwide are a major cause of morbidity and mortality.

When carrying out the POC diagnostic test method using LFIA, it was important to mitigate the spread of infectious diseases because it enabled rapid disease diagnosis and decision-making such as quarantine. LFIAs mainly use nanoparticles as signal amplifiers for the identification of target molecules by conjugating antibodies to nanoparticles, such as gold nanoparticles [2], carbon nanoparticles [3], dye-loaded latex beads [4], quantum dots [5], liposomes [6], magnetic nanoparticles [7] and up-converting phosphor nanoparticles [8].

β-diketonate Eu^3+^ complexes are widely used in electronic industry and biological applications due to their excellent luminescent properties, resistance to toxicity and good stability to photobleaching [9]. Photobleaching of organic dyes usually occurs within a few hours to a day and limits the spatial tracking of proteins or organelles in cells. Therefore, the β-diketonate Eu^3+^ complexes are one of the most preferred luminescent probes in the field of bioimaging [10]. LFIA with β-diketonate Eu^3+^ complexes enclosed in latex spheres have been widely applied because sensitivity can be improved hundreds of times or more compared to conventional gold-based LAFIA. Stable photophysical properties such as extended photoluminescence lifetime [11] and anti-photobleaching [12] allow the detection of numerous pathogens and inflammatory marker and can be used for early diagnosis of diseases [13].

However, it is impossible to directly use the actual β-diketonate Eu^3+^ complexes for diagnosis; these tend to be used by adding or chelating the Eu^3+^ complexes onto the surface of particle by adding them to the inside or outside of the polymer, metal or inorganic particles [14]. Incorporation of β-diketonate Eu^3+^ complexes into spherical hybrid composites enables photobleaching due to intramolecular energy transfer from antenna ligands bound to lanthanide metals and Eu^3+^ ions well shielded from the chemical environment [15].

High-energy oscillators such as C-H and O-H bonds contained in the β-diketone moiety non-radiatively interfere with the Eu^3+^ ions’ excited state, thereby suppressing the luminescence intensity and shortening the lifetime of the excited state. It is preferable to use β-diketone molecules containing C-F bonds instead of C-H bonds [16]. The ß-diketonate ligand of TFTB was selected because it possesses an electron-withdrawing -CF_3_ group as an antenna moiety for trivalent Eu^3+^ ions in order to absorb UV phonons and transfer the absorbed energy to the central Eu^3+^ ions [17]. The nonionic ligand of P(Oct)_3_ acts as a luminescent promotor as a component of the ternary europium complex.

In this work, highly monodispersed poly(St-*co*-MAA) nanoparticles containing β-diketonate Eu^3+^ complexes were synthesized using the step-wise process. The chemical structure of β-diketonate Eu^3+^ complexes of Eu(TFTB)_2_(MAA)P(Oct)_3_ and (b) core–shell structure of Eu(TFTB)_2_(MAA)P(Oct)_3_@poly(St-*co*-MAA) were illustrated in Figure 1a,b. Additionally, styrene and acrylic acid for latex monomers have similar luminescence quenching problems. When a cation or anion is exposed to the europium complex, the fluorescence properties is deteriorated by a substitution reaction between the cation and Eu^3+^ ions. MAA is introduced with four purposes: (i) acrylic groups for further polymerization with poly(St-*co*-MAA) nanoparticles, (ii) carboxylic acid group for complexation of Eu^3+^ ions, (iii) stabilizer to avoid dissociation from ambient conditions such as temperature and ionic impurities and (iv) hydrophilic moiety extended in the aqueous media.

The photochemical properties of the synthesized Eu(TFTB)_2_(MAA)P(Oct)_3_@poly(St-*co*-MAA) particles were investigated in order to determine whether particles could withstand photobleaching, cytotoxicity and fluorescence properties after intracellular uptake of particles in HepG2 cells.

## 2. Results and Discussion

### 2.1. Synthetic Schemes of Eu(TFTB)_2_(MAA)P(Oct)_3_@poly(St-co-MAA)

Figure 1a represents the chemical structure of β-diketonate Eu^3+^ complexes. Eu(TFTB)_2_(MAA)P(Oct)_3_ was chosen as a phosphor to construct europium encapsulated fluorescent latex nanoparticles because of its high PL properties and compatibility with monomers, initiator, and solvents. Figure 1b shows the core–shell structure of Eu(TFTB)_2_(MAA)P(Oct)_3_-coated poly(St-*co*-MAA) nanoparticles. The synthesis of Eu^3+^-complex modified latex particles has been reported in several methods. However, before designing the synthesis of Eu(TFTB)_2_(MAA)P(Oct)_3_@poly(St-*co*-MAA), the issue of luminescence quenching must be considered, as cationic and anionic surfactants can quench the luminescent properties of Eu^3+^ complexes. Figure 1c shows photograph images of poly(St-*co*-MAA) (left) and Eu(TFTB)_2_(MAA)P(Oct)_3_@poly(St-*co*-MAA) (right) dispersed in water were taken under (i) daylight, (ii) day and UV light (λ_ex_ at 365 nm) and (iii) UV light.

When MAA was used for copolymerization of poly(St-*co*-MAA) nanoparticles, acidity from carboxyl acid in MAA disturbed the formation of monodispersed latex nanoparticles; thus, NaOH and NaCO_3_ are involved in the regulation of pH and ionic strength, resulting in carboxylate ion during the synthesis of poly(St-*co*-MAA) nanoparticles. NaOH deprotonates the carboxyl group at the end of MAA. Without using NaOH and NaCO_3_, the synthesis conditions require higher temperatures and longer reaction times together with irregular size distribution. After forming poly(St-*co*-MAA) nanoparticles, sodium methacrylate on the surface of poly(St-*co*-MAA) nanoparticles was reactivated to carboxylic group by adding HCl. The activated carboxylic acid further chelated with Eu(TFTB)_2_(MAA)P(Oct)_3_ and polymerized between acrylic groups of poly(St-*co*-MAA) and Eu(TFTB)_2_(MAA)P(Oct)_3_ initiated with 2,2′-Azobis 2-methylpropionamidine dihydrochloride.

### 2.2. Characterizations of Eu(TFTB)_2_(MAA)P(Oct)_3_@poly(St-co-MAA)

The synthesized 2D-arranged poly(St-*co*-MAA) particles are spherical and have a uniform size distribution with a size of 288 nm, as shown in Figure 2a,b. The close packed arrangement of poly(St-*co*-MAA) are clear evidence that poly(St-*co*-MAA) particles have highly monodispersed size distribution because surface charge of the poly(St-*co*-MAA) particles are near to neutral. The narrow particle size distribution was due to the constant ionic strength of the polymerization medium. Figure 2c,d represents poly(St-*co*-MAA) particles after chelating with Eu(TFTB)_2_(MAA)P(Oct)_3_. The as synthesized poly(St-*co*-MAA) particles were postmodified with HCl to recover the carboxylic group for the further chelation with Eu^3+^ complex. Figure 2e shows particle size distributions of poly(St-*co*-MAA) and Eu(TFTB)_2_(MAA)P(Oct)_3_@poly(St-*co*-MAA) nanoparticles. Particles size of poly(St-*co*-MAA) was increased to 312 nm after chelating with Eu(TFTB)_2_(MAA)P(Oct)_3_. No applicable distinct shape changes were also monitored after modification with Eu(TFTB)_2_(MAA)P(Oct)_3_@poly(St-*co*-MAA). However, the particles no longer form a 2D structural arrangement due to the change in the surface charge. After coating with the Eu(TFTB)_2_(MAA)P(Oct)_3_ complex, it did not undergo significant aggregation and agglomeration, which significantly affected the colloidal stability of poly(St-*co*-MAA) in an aqueous medium.

The characteristic thermal behaviors of poly(St-*co*-MAA) and Eu(TFTB)_2_(MAA)P(Oct)_3_@poly(St-*co*-MAA) were analyzed by DSC, and the chelated amount of Eu(TFTB)_2_(MAA)P(Oct)_3_ on poly(St-*co*-MAA) was estimated by TGA (Figure 3) [18]. The TGA profiles for poly(St-*co*-MAA) and Eu(TFTB)_2_(MAA)P(Oct)_3_@poly(St-*co*-MAA) show weight loss of about 82% and 93.3% up to 430 °C, respectively. The weight loss difference of 5.3% indicates the amount of chelated β-diketonate Eu^3+^ complexes on the poly(St-*co*-MAA) surface. The weight change in the heating range from room temperature to 120 °C is due to the dehydration of physically adsorbed moisture to the particles.

The secondary weight loss and heat flow curve of Eu(TFTB)_2_(MAA)P(Oct)_3_@poly(St-*co*-MAA) over the temperature range of 200 °C to 527 °C is due to the thermal decomposition of organic compounds such as MAA, TFTB and P(Oct)_3_. The heat of the reaction increased due to the disintegration of the β-diketonate Eu^3+^ complexes on poly(St-*co*-MAA). Eu(TFTB)_2_(MAA)P(Oct)_3_@poly(St-*co*-MAA) has a lower decomposition temperature than poly(St-*co*-MAA), which means better thermal conductivity [19]. The sharp endothermic peak in the DSC curve of Eu(TFTB)_2_(MAA)P(Oct)_3_@poly(St-*co*-MAA) at 430 °C can be assigned to the thermal decomposition of P(Oct)_3_.

Figure 4 shows FTIR spectra of poly(St-*co*-MAA), Eu(TFTB)_2_(MAA)P(Oct)_3_ and Eu(TFTB)_2_(MAA)P(Oct)_3_@poly(St-*co*-MAA). The FTIR spectrum of Eu(TFTB)_2_(MAA)P(Oct)_3_ shows the C–P stretching peaks of P(Oct)_3_ appeared at 1083 and 1060cm^−1^. The peak at 714 cm^−1^ related to –CH_2_ stretching of P(Oct)_3_ was also observed. Moreover, the peaks at 1497, 1464 and 1381 cm^−1^ coming from the asymmetric in-plane and symmetric rocking mode of the terminal methyl group of P (Oct)_3_.

The peaks at 1413 and 1354 cm^−1^ can be assigned to ν (C=C, C=S). CF_3_ appeared in TFTB vibration peaks at 1302 cm^−1^ (ν(CF_3_)) and 716 cm^−1^ (δ(CF_3_)). The peak at 1536 cm^−1^ (ν (C=O)) can be assigned to the vibration peak of the keto-enol tautomerization of β-diketone. The peaks at 1060 cm^−1^ (δ(O-CH_3_)) and 857 cm^−1^ (δ(CH_3_)). The peaks at 1695 cm^−1^, 1634 cm^−1^, 1416 cm^−1^, and 1301 cm^−1^ correspond to the levels of C=O, C=C, –CH_2_ and –CH_3_ coming from MAA.

Figure 5 shows XPS survey profiles (Figure 5a) and individual elements in Eu(TFTB)_2_(MAA)P(Oct)_3_@poly(St-*co*-MAA). A strong signal from the C_1s_ element is expected from poly(St-*co*-MAA) and β-diketonate Eu^3+^ complexes. Figure 5b represents the maximum position at 1135.5 eV (0.04 at%) coming from the Eu_3d_ core-level spectrum of the β-diketonate Eu^3+^ complexes. The weak peak at 688.3 eV (0.51 at%) comes from the F1s core-level spectrum, which can be assigned to -CF_3_ in TFTB (Figure 5c). The strong peak at 532 eV (3.23 at%) indicates the presence of the core-level spectrum of O_1s_ due to the carboxylic acid in MAA. Figure 5d,e shows the position of the maximum peak at 168.5 eV (0.56 at%) coming from the F_1s_ core-level spectrum due to -CF_3_ in TFTB. The peak appeared at 133 eV (0.52 at%) from the P_2p_ core-level spectrum (Figure 5f).

Figure 6 displays the emission (*E_m_*) and excitation (*E_x_*) spectra, depending on different concentrations of Eu(TFTB)_2_(MAA)P(Oct)_3_ and Eu(TFTB)_2_(MAA)P(Oct)_3_@poly(St-*co*-MAA) nanoparticles dispersed in water. The photoluminescence (PL) spectrum of the β-diketonate Eu^3+^ complex provides more photochemical information than the corresponding UV absorption spectrum. In general, β-diketonate Eu^3+^ complexes exhibit a strong PL because of ^5^D_0_→^7^F_J(J = 0–6)_ transition [19,20,21]. Figure 6a is the *E_m_* spectrum (λ_ex_ at 350 nm) of Eu(TFTB)_2_(MAA)P(Oct)_3_ dispersed in water. The *E_m_* spectra of Eu(TFTB)_2_(MAA)P(Oct)_3_ consist of multiple *E_m_* peaks at around 585, 597, 612 and 650 nm, which are assigned to ^5^D_0_→^7^F_J(J = 0–3)_ transitions of Eu^3+^, respectively. The f–f intraconfigurational transition lines were assigned to ^5^D_0_→^7^F_0-4_ dominated by the hypersensitive ^5^D_0_→^7^F_2_ transition [22]. ^5^D_0_→^7^F_0_ occurs because Eu^3+^ ions of Eu(TFTB)_2_(MAA)P(Oct)_3_ occupy a position with low symmetry [23], whereas ^5^D_0_→^7^F_1_ is associated with magnetic-dipole transition, regardless of the surrounding Eu^3+^. It is well known that the ^5^D_0_→^7^F_0,1_ transition is directed by selection rules for the intermediate magnetic-dipole coupling of ∆J = 0, ±1, and that ^5^D_0_→^7^F_2,4,6_ allows for electro-dipole transitions [24].

Moreover, ^5^D_0_→^7^F_3_ includes both magnetic dipole and electric transition, and the ^5^D_0_→^7^F_4_ is related to the interaction of an electron in europium atom with the electromagnetic radiation. By comparing the intensity ratio (^5^D_0_→^7^F_2_/^5^D_0_→^7^F_1_), it can be used as a relative indicator of Eu^3+^ exposed to the surrounding environment [25]. In this study, the ^5^D_0_→^7^F_2_/^5^D_0_→^7^F_1_ level is about 9.4, which implies a low symmetrical coordination near to the Eu^3+^ ions [26]. Figure 6b shows that the Ex spectrum (λ_em_ = 621 nm) of Eu(TFTB)_2_(MAA)P(Oct)_3_ is recognized a maximum *E_x_* peak at 352 nm [27]. Figure 6c shows the maximum *E_m_* peak at 621 nm, which indicates a strong red *E_m_* characteristic associated with the electric dipole ^5^D_0_→^7^F_2_ transition of Eu(TFTB)_2_(MAA)P(Oct)_3_ [28]. Figure 6d shows the *E_x_* spectrum (λ_em_ = 621 nm) of Eu(TFTB)_2_(MAA)P(Oct)_3_@poly(St-*co*-MAA) is recognized a sharp peak at 309 nm.

Figure 7a–c show the luminescent decay of the emission state (^5^D_0_) of measured upon monitoring the hypersensitive ^5^D_0_→^7^F_2_ transition. The PL lifetimes (τ) were determined to be 284 ns for Eu(TFTB)_2_(MAA)P(Oct)_3_ at 375 nm, and 370 ns for Eu(TFTB)_2_(MAA)P(Oct)_3_@poly(St-*co*-MAA) at 375 nm. Lifetime of Eu(TFTB)_2_(MAA)P(Oct)_3_ was decreased after coating on the poly(St-*co*-MAA). An increase in fluorescence is associated with a decrease in lifetime. However, the fluorescence intensity does not increase linearly with lifetime. The lifetime of a fluorescent material refers to the average time a particle remains in an excited state, and the fluorescence intensity is related to the photon emitted from the ligand, which is absorbed by the antenna ligand and then transferred to the lanthanide ion [29].

### 2.3. Cytotoxicity of Poly(St-co-MAA) and Eu(TFTB)_2_(MAA)P(Oct)_3_ @poly(St-co-MAA)

Viability assay is critical procedure in nanotoxicology that describe the cellular response to nanoparticulate materials. Moreover, it gives information on cytotoxicity, survival rate, and metabolic activities. For a systematic and comparative study on the cytotoxic effects of poly(St-*co*-MAA) and Eu(TFTB)_2_(MAA)P(Oct)_3_@poly(St-*co*-MAA), human hepatic cell (HepG2) line was used to study the cell viability according to the concentration-dependent and time-dependent cellular uptake of nanoparticles. Cell viability was performed by colorimetric WST-8 assay (450 nm), which involved n = 96 wells within N = 3 independent technical repeats, as described in the experimental section. Cell viability for poly(St-*co*-MAA) and Eu(TFTB)_2_(MAA)P(Oct)_3_@poly(St-*co*-MAA) were normalized to the viability of medium as a control and are shown as percentages of the medium control in Figure 8.

For the concentration-dependent data at 24 h, cell viability was 96.1 ± 2.75% for 20 μg/mL, 95.5 ± 2.7% for 50 μg/mL, 95.2 ± 4.5% for 100 μg/mL and 99.0 ± 1.5% for 250 μg/mL of poly(St-*co*-MAA)-treated HepG2 cells and 104.5 ± 9.2% for 20 μg/mL, 98.9 ± 8.8% for 50 μg/mL, 103.0 ± 7.4% for 100 μg/mL and 112.0 ± 6.5% for 250 μg/mL of Eu(TFTB)_2_(MAA)P(Oct)_3_@poly(St-*co*-MAA) treated HepG2 cells. At a concentration of 100 µg/mL, cell viability was 102.1 ± 2.7% for 3 h, 103.0 ± 1.6% for 6 h, 100.7 ± 5.6% for 9 h, 101.5 ± 2.3% for 12 h, 103.6 ± 5.7% for 24 h and 98.9 ± 8.8% for 48 h of poly(St-*co*-MAA) treated HepG2 cells and 108.9 ± 8.9% for 3 h, 104.6 ± 8.2% for 6 h, 101.1 ± 8.3% for 9 h, 107.5 ± 6.4% for 12 h 100.8 ± 8.8% for 24 h and 97.2 ± 7.6% for 48 h of Eu(TFTB)_2_(MAA)P(Oct)_3_@poly(St-*co*-MAA) treated HepG2 cells. Compared with the control group, the viability of HepG2 cells exposed to Eu(TFTB)_2_(MAA)P(Oct)_3_@poly(St-*co*-MAA) was not significantly different despite the increase in concentration from 20 to 250 µg/mL. At the dose treated as described above, poly(St-*co*-MAA) did not show a decrease in cell viability, and there was no significant change in cytotoxicity when cells were treated and cultured at 100 µg/mL for 24 h [30].

### 2.4. Endocytosis of Poly(St-co-MAA) and Eu(TFTB)_2_(MAA)P(Oct)_3_@poly(St-co-MAA) Nanoparticles by Flow Cytometry

Flow cytometry assays for necrotic/apoptotic cells have benefits over traditional viability assays such as annexin V/PI, which do not provide quantitative information on dead cells. Flow cytometry facilitates accurate quantification of live, early apoptotic, late apoptotic and necrotic cells in large cell populations [31]. Concentration dependent cell viability was investigated after treating the HepG2 cells with poly(St-*co*-MAA) and Eu(TFTB)_2_(MAA)P(Oct)_3_@poly(St-*co*-MAA) and was measured by flow-cytometry-based assays. After treating the cells, Annexin-V/PI was stained as a marker for early apoptosis. Based upon the degree of uptake of Annexin-V/PI, the population of individual cells are plotted in one of the four quadrants. The cells without staining with Annexin-V are identified as non-apoptotic, non-necrotic cells (live cells). The cells stained for Annexin-V dye are early apoptotic cells (Figure 9). The synthesized poly(St-*co*-MAA) and Eu(TFTB)_2_(MAA)P(Oct)_3_@poly(St-*co*-MAA) nanoparticles were confirmed to be virtually non-toxic in terms of cytotoxicity in HepG2 cells. After 24 h of incubation, apoptosis in late-stage apoptosis were 1.48% ± 0.2% for 20 µg/mL, 1.08% ± 0.0% for 50 µg/mL, 1.20% ± 0.1% for 100 µg/mL, 1.31% ± 0.1% for 250 µg/mL of poly(St-*co*-MAA) treated HepG2 cells, respectively, and 1.08% ± 0.7% for 20 µg /mL, 1.52% ± 0.2% for 50 µg/mL, 0.84% ± 0.3% for 100 µg/mL, 0.91% ± 0.7% for 250 µg/mL of Eu(TFTB)_2_(MAA)P(Oct)_3_@poly(St-*co*-MAA) treated HepG2 cells, respectively. Consequently, cell-specific fluorescence of Eu(TFTB)_2_(MAA)P(Oct)_3_@poly(St-*co*-MAA) indicated that the particle dynamics of Eu(TFTB)_2_(MAA)P(Oct)_3_@poly(St-*co*-MAA) undergo endocytosis. There are fewer early apoptotic, late apoptotic and necrotic cells in each sample, regardless of culture period compared with live cells.

### 2.5. Intracellular Uptake of Poly(St-co-MAA) and Eu(TFTB)_2_(MAA)P(Oct)_3_@poly(St-co-MAA) Nanoparticles

Because of their small size, nanoparticles can easily enter cells as well as migrate across cells. Nanoparticles are widely used in nanomedicinal applications since they form complex with biomolecules and overcome the cell plasma membrane and cytoplasmic transport to achieve high therapeutic efficacy. HepG2 cells were incubated with poly(St-*co*-MAA) and Eu(TFTB)_2_(MAA)P(Oct)_3_@poly(St-*co*-MAA) nanoparticles for predetermined time and the nuclei were stained with DAPI (blue). As shown in Figure 10, Eu(TFTB)_2_(MAA)P(Oct)_3_@poly(St-*co*-MAA) particles showed a stronger red fluorescence signals than control and poly(St-*co*-MAA), which implied that Eu(TFTB)_2_(MAA)P(Oct)_3_@poly(St-*co*-MAA) were actively trafficked within cells. The individual nanoparticles present in early endosomes were gradually clustered via vesicle fusion during the maturation process [32]. The small clusters were randomly distributed in the cytoplasm, whereas large clusters were mostly localized to the perinuclear region. The strong and large red emitting dots in the images could be explained by the accumulation or agglomeration of Eu(TFTB)_2_(MAA)P(Oct)_3_@poly(St-*co*-MAA) in the lysosome of the cells. In addition, Eu(TFTB)_2_(MAA)P(Oct)_3_@poly(St-*co*-MAA) synthesized in this work can be excited in the full UV range with a maximum *E_m_* at 619 nm. Additionally, these particles can used for red luminescent organic dyes, lateral flow detection, latex agglutination tests, markers to label biomolecules, and intracellular trafficking and cellular imaging agents.

## 3. Materials and Methods

### 3.1. Materials

Styrene (St) was supplied by Junsei Chemicals (Tokyo, Japan). Europium chloride hexahydrate, trioctylphosphine (P (Oct)_3_), 2,2’-Azobis 2-methylpropionamidine dihydrochloride and dialysis tubing cellulose membrane (Mw cut-off = 14,400) membrane were purchased from Sigma-Aldrich. Methacrylic acid (MAA) was supplied by Yakuri Pure Chemicals Co., Ltd. (Kyoto, Japan). Potassium persulfate (PPS), ethanol (EtOH), sodium hydroxide (NaOH), hydrochloric acid (HCl) and sodium carbonate (NaCO_3_) were supplied by Samchun Chemicals (Kyungkido, Republic of Korea)

4,4,4-Trifluoro-1-(2-thienyl)-1,3-butanedione (TFTB) was supplied by Tokyo Chemical Industry (Tokyo, Japan). Polyvinylpyrrolidone (PVP, Mw = 30,000) was purchased from Kanto Chemical (Tokyo, Japan). All of the chemicals were analytical grade and were used as received without further purification when it was not specially mentioned. Water was purified using the ELGA Flex 3 water purification system (VWR Co. Ltd., PA, USA).

### 3.2. Characterization

The morphology and particle size were analyzed by scanning electron microscope (SEM, SNE-4500M, SEC) at an operation voltage of 10 kV. A thin layer of gold was deposited on the sample using sputtering machine for 30 sec. Fourier transform infrared (FTIR) spectroscopy were collected directly by dropping the sample on the facet of diamond with an attenuated total reflection (ATR) mode. IR spectra were recorded using a Bruker ALPHA FT-IR (Platinum ATR with diamond module) with 256 scans at a resolution of 1 cm^−1^ and a wavenumber range of 400~4000 cm^−1^.

Differential scanning calorimetry (DSC) and thermal gravimetric analysis (TGA) were performed using STA-6000 (PerkinElmer, MA, USA) to measure the heat of fusion, specific heat and melting point depression of poly(St-*co*-MAA) and Eu(TFTB)_2_(MAA)P(Oct)_3_@poly(St-*co*-MAA) particles. About 30 mg of the sample was placed in an alumina fan and then cooled to the target temperature with a series of cooling rates, kept at an isothermal state for 10 min and heated to 700 °C at a rate of 10 °C/min. The photoluminescent (PL) properties of Eu(TFTB)_2_(MAA)P(Oct)_3_ and Eu(TFTB)_2_(MAA)P(Oct)_3_@poly(St-*co*-MAA) were investigated using RF-5301PC (Shimadzu, Japan) equipped with a 150 W xenon lamp. The chemical compositions of Eu(TFTB)_2_(MAA)P(Oct)_3_ and Eu(TFTB)_2_(MAA)P(Oct)_3_@poly(St-*co*-MAA) were characterized by XPS using a commercial VG Microtech Multilab ESCA 2000 with a CLAM MCD detector and Al Kα radiation (1486.6 eV), operating at 1 × 10^−8^ Torr. Irradiation survey scans were obtained in the range of 0–1400 eV with an energy step of 1.0 eV and pass energy of 100 eV. Luminescence attenuation was recorded in the 200–980 nm emission range at room temperature using a Horiba-Jobin-Yvon Fluorolog-QM spectrofluorometer equipped with a 75 W ArcTune xenon lamp and a Hamamatsu R-FL-QM-R13456 photomultiplier tube, and Hamamatsu QM-H10330. The -75-NIRTCSPC photomultiplier tube is sensitive when in the 950–1700 nm emission range.

### 3.3. Synthesis of Europium Complex Coated Poly(St-co-MAA) Nanoparticles, Eu(TFTB)_2_(MAA)P(Oct)_3_@poly(St-co-MAA)

Eu(TFTB)_2_(MAA)P(Oct)_3_ was synthesized according to a previously reported method. Briefly, stock solutions of Eu(III), P(Oct)_3_ and TFTB were prepared by dissolving them in EtOH. In total, 0.5 mmol Eu(III), 0.5 mmol P(Oct)_3_, 0.5 mmol MAA and 1 mmol TFTB were transferred to 100 mL round bottom flask and sealed tightly. The mixture solution was immersed in a water bath and heated at 65 °C for 1 h. The synthesized Eu(TFTB)_2_(MAA)P(Oct)_3_ in EtOH was kept at room temperature until use.

Monodispersed polystyrene (poly(St-*co*-MAA)) nanoparticles were synthesized using emulsifier-free polymerization initiated by PPS. Overall, 50 mL styrene was purified by the addition of 1g NaOH overnight. The inhibitor-free styrene was transferred to a clean vial and kept at 4 °C until use. A 100 mL 3 neck round-bottom flask was equipped with a condenser, a thermocouple, an oval magnetic bar and a nitrogen inlet. The flask was charged with 80 mL H_2_O and degassed under N_2_ for 30 min. A mixture of 7 g styrene (St) and 0.35 g methacrylic acid (MAA) was added to the flask at a stirring speed of 500 rpm. Total amounts of 0.024 g NaOH and 0.024 g NaCO_3_ were dissolved separately in 5 mL H_2_O, then poured into a flask and heated to 75 °C in an oil bath. After reaching the reaction temperature, 0.03g PPS dissolved in 5 mL H_2_O was added, deaerated under N_2_ for 10 min and heated in an oil bath at 75 °C for 12 h. A total of 1 mL 12 M HCl was added and stirred for 1 h to activate the carboxylic acid. The synthesized poly(St-*co*-MAA) nanoparticles were purified using a dialysis tube cellulose membrane against 5 L H_2_O for 2 days. The conductivity of the counterpart H_2_O was monitored to determine the terminal point of dialysis.

In total, 1 g poly(St-*co*-MAA) dispersed in 90 mL H_2_O was transferred to a 100 mL round-bottom flask. Then, 10 mL 0.1 mmol Eu(TFTB)_2_(MAA)P(Oct)_3_ dissolved in 10 mL EtOH and 0.1 g PVP and 0.025g 2,2’-Azobis 2-methylpropionamidine dihydrochloride dissolved in 10 mL H_2_O were added and sealed with a quick-fit glass stopper. The mixture was stirred using a magnetic stirrer at 25 °C with a stirring rate of 500 rpm for 24 h. Subsequently, the quick-fit glass stopper was removed, and the reaction was further continued to remove EtOH under magnetic stirring at 25 °C with a stirring rate of 500 rpm for 24 h. The as-synthesized Eu(TFTB)_2_(MAA)P(Oct)_3_@poly(St-*co*-MAA) nanoparticles were purified using dialysis tubing cellulose membrane against 5 L H_2_O for 2 days.

### 3.4. Immunofluorescence

HepG2 cells were seeded onto the slides and placed in the 24-well plates (4 × 10^5^ cells/well) and then the cells reach about 90% confluence. The HepG2 cells were cultured with poly(St-*co*-MAA) and Eu(TFTB)_2_(MAA)P(Oct)_3_@poly(St-*co*-MAA) at different concentrations and time periods and cultured at 37 °C in a humidified 5% CO_2_ environment. Coverslips were rinsed once with 1 × PBS and water. The coverslips were carefully removed from the wells and fixed with 4% paraformaldehyde. The coverslips were mounted with 4′,6-diamidino-2-phenylindole (DAPI, Invitrogen). Samples were analyzed using an epifluorescence microscope (DM2500, Leica, Wetzlar, Germany).

### 3.5. Cytotoxicity Assay

Cell viability assay was determined using the WST-8 assay (Daeil Lab Service Co., Seoul, Republic of Korea). Briefly, HepG2 cells at a density of 1 × 10^4^ cells per well were plated in 96-well plates and treated at different concentrations of poly(St-*co*-MAA) and Eu(TFTB)_2_(MAA)P(Oct)_3_@poly(St-*co*-MAA) cultured for 24 h. Subsequently, The WST-8 reagent was incubated for 30 min at 37 °C in a 5% CO_2_ incubator (Thermo Fisher Scientific Inc., Seoul, Republic of Korea). Absorbance was automatically measured at 450 nm with a microplate reader (BioTek Instruments Korea Ltd., Seoul, Republic of Korea). The percentage of cell viability was expressed as a percentage of the optical density between treated cells and the control group.

### 3.6. Apoptosis Assay

HepG2 cells at a density of 4 × 10^5^ cells per well were seeded in 12-well plates for 12 h. After incubation with poly(St-*co*-MAA) and Eu(TFTB)_2_(MAA)P(Oct)_3_@poly(St-*co*-MAA) at indicated concentration for 24 h, adherent and floating cells were collected in a 1.5 mL tube through trypsinization, washed with PBS, and stained using Annexin V-FITC/PI apoptosis detection kit (Thermo Fisher Scientific, Inc., Seoul, Republic of Korea). The percentage of apoptotic cells was quantified using flow cytometer (Beckman Coulter, Fullerton, CA, USA).

### 3.7. Statistical Analysis

Unless otherwise stated, results were presented as mean ± standard error of the mean (SEM), and were representative obtained from at least three independent experiments. Student’s t-test was used to compare the means between the two groups. A *p*-value less than 0.05 (typically ≤ 0.05) is statistically significant.

## 4. Conclusions

In this work, we synthesized Eu(TFTB)_2_(MAA)P(Oct)_3_@poly(St-*co*-MAA) composed of TFTB, MAA, P(Oct)_3_ and Eu(III). In particular, MAA was introduced into both the europium complex and polystyrene particles. Acrylic groups of MAA were used for further polymerization between poly(St-*co*-MAA) and europium complexes, and carboxylic acids were reacted with the Eu^3+^ ions. MMA acts as a stabilizer to prevent dissociation of weakly bound europium complexes that are affected by ambient conditions such as temperature and ionic impurities, etc. Additionally, the hydrophilic nature of the carboxylic acids in MMA markedly enhances the dispersion of Eu(TFTB)_2_(MAA)P(Oct)_3_@poly(St-*co*-MAA) in aqueous media.

The synthesized poly(St-*co*-MAA) particles are spherical with a size distribution of 300 nm. The close packed arrangement of poly(St-*co*-MAA) are clear evidence that poly(St-*co*-MAA) particles have a highly monodispersed size distribution because the surface charge of the poly(St-*co*-MAA) particles is near to neutral. The *E_m_* spectra of Eu(TFTB)_2_(MAA)P(Oct)_3_ consist of multiple bands of *E_m_* at around 585, 597, 612 and 650 nm, which is assigned to ^5^D_0_→^7^F_J(J = 0–3)_ transitions of Eu^3+^, respectively. The f–f intraconfigurational transition lines were assigned to ^5^D_0_→^7^F_0-4_ dominated by the hypersensitive ^5^D_0_→^7^F_2_ transition the ^5^D_0_→^7^F_2_/^5^D_0_→^7^F_1_ is about 9.4, which implies a low symmetrical coordination near Eu^3+^ ions. The maximum *E_m_* peak at 621 nm, which indicates a strong red *E_m_* characteristic associated with the electric dipole ^5^D_0_→^7^F_2_ transition of Eu(TFTB)_2_(MAA)P(Oct)_3_.

For a systematic and comparative study on the cytotoxic effects of poly(St-*co*-MAA) and Eu(TFTB)_2_(MAA)P(Oct)_3_@poly(St-*co*-MAA), human hepatic cell (HepG2) line was used to study the cell viability study according to the concentration-dependent and time-dependent cellular uptake of nanoparticles. At the dose treated as described above, poly(St-*co*-MAA) did not show a decrease in cell viability, and there was no significant change in cytotoxicity when cells were treated and cultured at 100 μg/mL for 24 h. As expected, the compositional moieties of the europium complex, TFTB, MAA and P(Oct)_3_, have very low cytotoxicity, their chelation compounds also exhibit very high cell viability. Consequently, cell-specific fluorescence of Eu(TFTB)_2_(MAA)P(Oct)_3_@poly(St-*co*-MAA) indicated the particle dynamics of Eu(TFTB)_2_(MAA)P(Oct)_3_@poly(St-*co*-MAA) undergoes endocytosis. There are fewer early apoptotic, late apoptotic and necrotic cells in each sample regardless of culture period compared to live cells.

These results suggest that highly monodisperse Eu(TFTB)_2_(MAA)P(Oct)_3_@poly(St-*co*-MAA) nanoparticles prepared in this work have an *E_m_* band in the entire UV range with a maximum *E_m_* at 621 nm. Additionally, these particles can replace red emitting organic/inorganic dyes. Therefore, Eu(TFTB)_2_(MAA)P(Oct)_3_@poly(St-*co*-MAA) can be used for bioimaging, luminescent sensors, as a probe for clinical use, and for intracellular trafficking and cellular imaging agents.

## Figures and Tables

**Figure 1 ijms-23-15954-f001:**
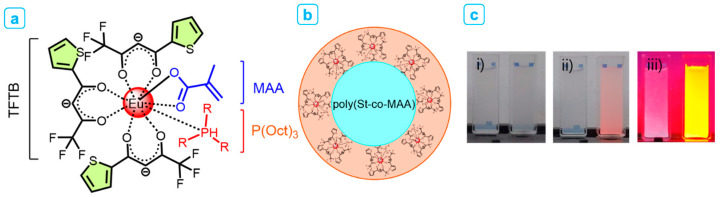
Schematic representation of (**a**) chemical structure of β-diketonate Eu^3+^ complexes of Eu(TFTB)_2_(MAA)P(Oct)_3_ and (**b**) core–shell structure of Eu(TFTB)_2_(MAA)P(Oct)_3_@poly(St-*co*-MAA). (**c**) photograph images of poly(St-*co*-MAA) (left) and Eu(TFTB)_2_(MAA)P(Oct)_3_@poly(St-*co*-MAA) (right) dispersed in water were taken under (i) daylight, (ii) day and UV light (λ_ex_ at 365 nm) and (iii) UV light.

**Figure 2 ijms-23-15954-f002:**
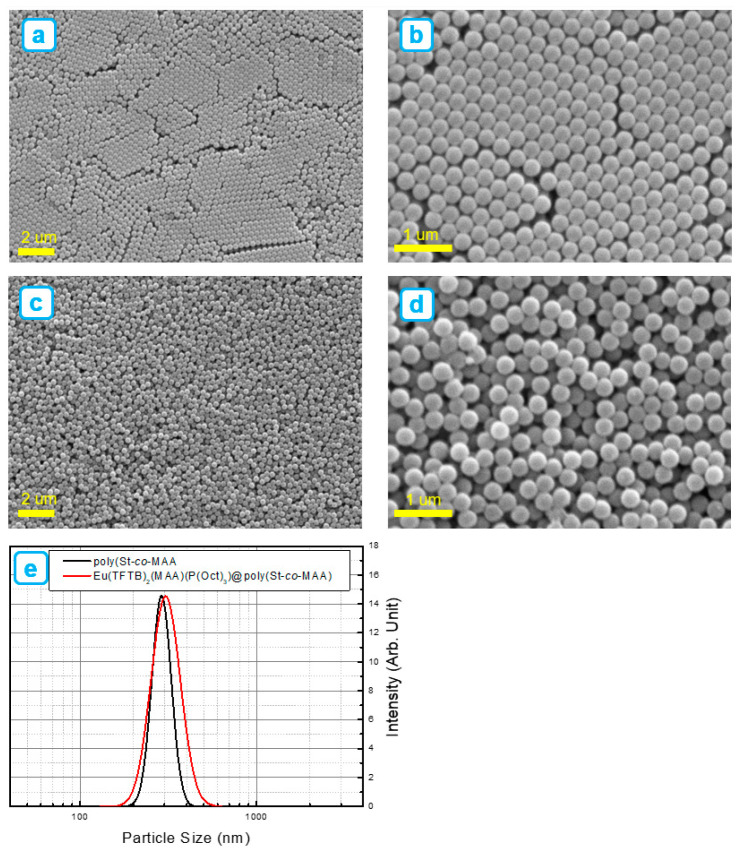
SEM images of synthesized: (**a**,**b**) polystyrene (poly(St-*co*-MAA) nanoparticles, (**c**,**d**) after modifying poly(St-*co*-MAA) nanoparticles with β-diketonate Eu^3+^ complexes of Eu(TFTB)_2_(MAA)P(Oct)_3_ resulting in Eu(TFTB)_2_(MAA)P(Oct)_3_@poly(St-*co*-MAA). (**e**) Particle size distributions of poly(St-*co*-MAA) and Eu(TFTB)_2_(MAA)P(Oct)_3_@poly(St-*co*-MAA) nanoparticles.

**Figure 3 ijms-23-15954-f003:**
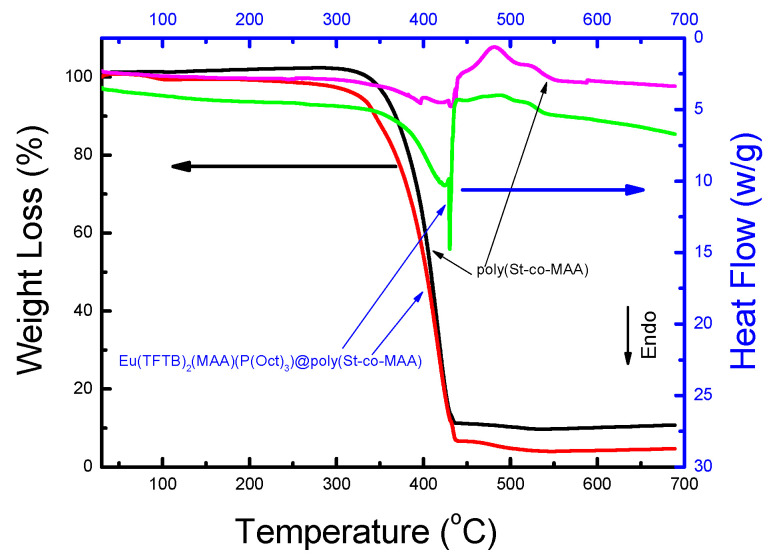
Thermal degradation profile and DSC curve of poly(St-*co*-MAA) and Eu(TFTB)_2_(MAA)P(Oct)_3_@poly(St-*co*-MAA) particles.

**Figure 4 ijms-23-15954-f004:**
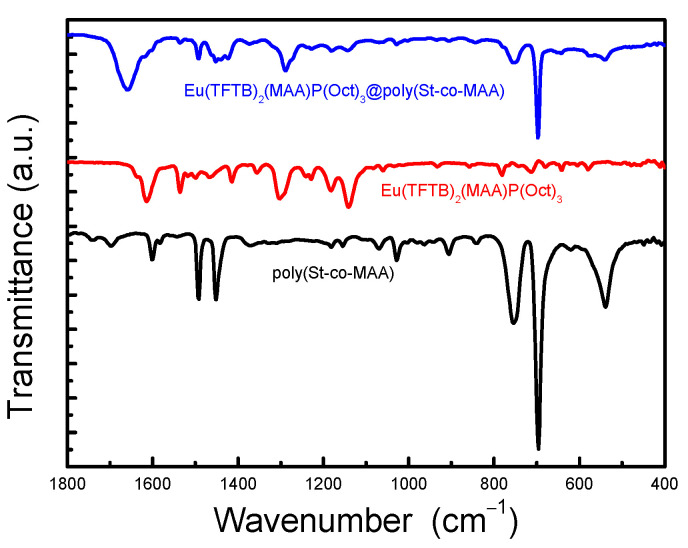
FTIR transmittance spectra of poly(St-*co*-MAA), Eu(TFTB)_2_(MAA)P(Oct)_3_ and Eu(TFTB)_2_(MAA)P(Oct)_3_@poly(St-*co*-MAA).

**Figure 5 ijms-23-15954-f005:**
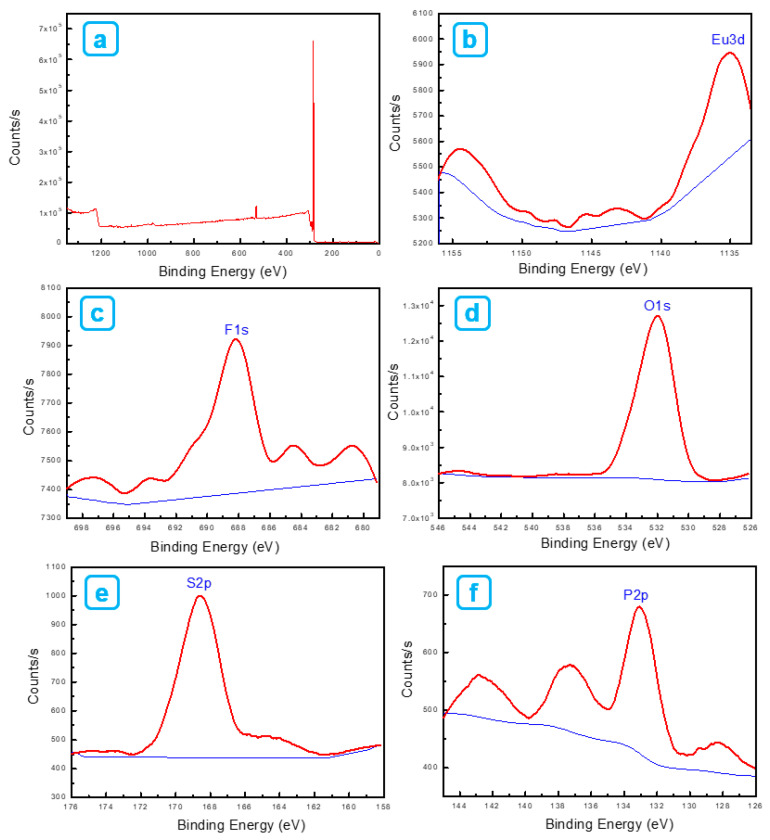
(**a**) X-ray photoelectron spectroscopy survey spectra of Eu(TFTB)_2_(MAA)P(Oct)_3_@poly(St-*co*-MAA). (**b**) Eu_3d_ core-level spectrum, (**c**) F_1s_ core-level spectrum, (**d**) O_1s_ core-level spectrum, (**e**) S_2p_ core-level spectrum and (**f**) P_2p_ core-level spectrum of Eu(TFTB)_2_(MAA)P(Oct)_3_@poly(St-*co*-MAA).

**Figure 6 ijms-23-15954-f006:**
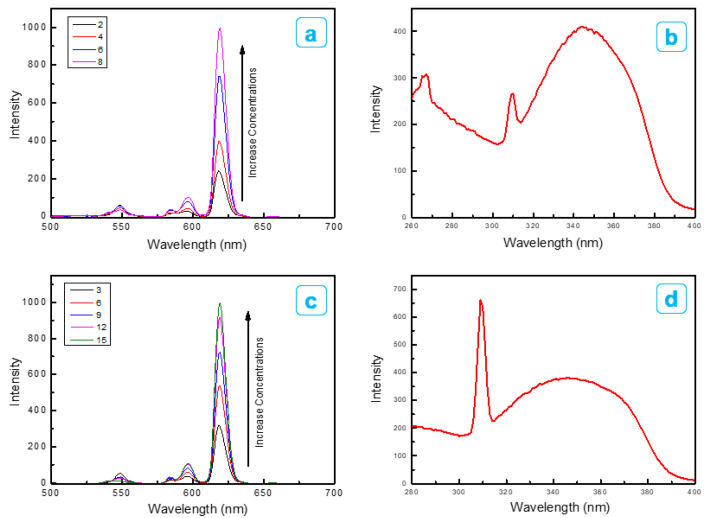
(**a**) Emission spectrum (λ_ex_ at 350 nm) and (**b**) excitation spectrum (λ_em_ at 621 nm) Eu(TFTB)_2_(MAA)P(Oct)_3_ dispersed in water and (**c**) Emission spectra (λ_ex_ at 342 nm) and (**d**) excitation spectrum (λ_em_ at 621 nm) at 298 K for ^5^D_0_→^7^D_j_ transition depending on different concentration of Eu(TFTB)_2_(MAA)P(Oct)_3_@poly(St-*co*-MAA) dispersed in water.

**Figure 7 ijms-23-15954-f007:**
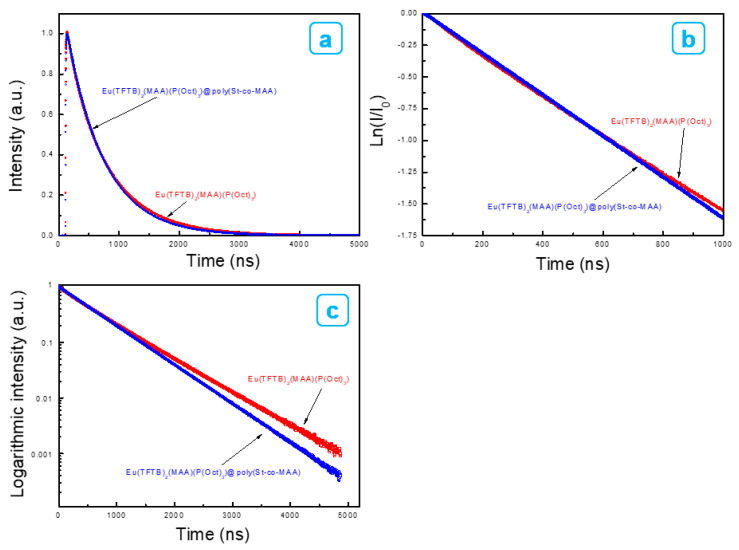
Luminescence decay curves for Eu(TFTB)_2_(MAA)(P(Oct)_3_) (red) and Eu(TFTB)_2_(MAA)(P(Oct)_3_)@poly(St-*co*-MAA) (blue) dispersed in water (λ_em_ = 615 nm, λ_ex_ = 375 nm). Plotted with (**a**) time (ns) versus intensity, (**b**) time (ns) versus natural logarithm (I/I_0_) and (**c**) time (ns) versus logarithmic intensity.

**Figure 8 ijms-23-15954-f008:**
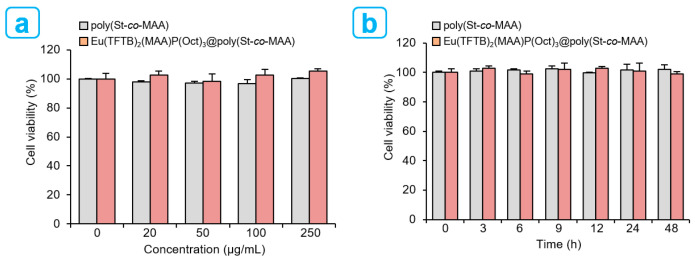
Cell viability of HepG2 cells treated with poly(St-co-MAA) and Eu(TFTB)_2_(MAA)P(Oct)_3_@poly(St-co-MAA) by WST-8 assay. (**a**) Concentration-dependent cell viability of HepG2 cells treated with control and 20, 50, 100 and 250 μg/mL of poly(St-co-MAA) and Eu(TFTB)_2_(MAA)P(Oct)_3_@poly(St-co-MAA) dispersed in PBS for 24 h. (**b**) Time-dependent cell viability of HepG2 cells treated with 100 µg/mL of poly(St-co-MAA) and Eu(TFTB)_2_(MAA)P(Oct)_3_@poly(St-co-MAA) for different incubation times of 0, 3, 6, 9, 12, 24 and 48 h.

**Figure 9 ijms-23-15954-f009:**
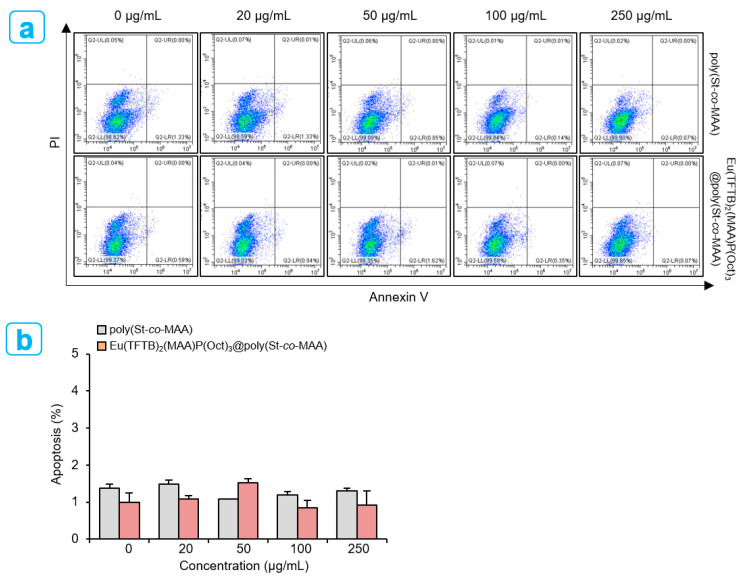
(**a**) Concentration-dependent cell viability of HepG2 cells treated with 20, 50, 100 and 250 µg/mL of poly(St-*co*-MAA) and Eu(TFTB)_2_(MAA)P(Oct)_3_@poly(St-*co*-MAA) nanoparticles for 24 h. The cells were stained with Annexin V-FITC and quantified for apoptosis by flow cytometer. Top right quadrant, dead cells in late stage of apoptosis; bottom right quadrant, cells undergoing apoptosis; bottom left quadrant, viable cells. (**b**) The rate of apoptosis depends on the concentration of poly(St-*co*-MAA) and Eu(TFTB)_2_(MAA)P(Oct)_3_@poly(St-*co*-MAA) nanoparticles. Data are shown as means ± SEM. Results are representative of at least three independent experiments.

**Figure 10 ijms-23-15954-f010:**
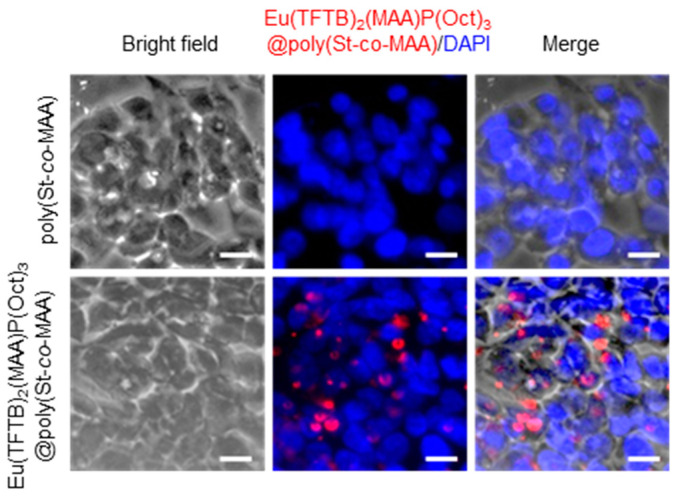
Fluorescence images of intracellular uptake and trafficking of poly(St-*co*-MAA) and Eu(TFTB)_2_(MAA)P(Oct)_3_@poly(St-*co*-MAA) in HepG2 cells. Nuclei were stained with DAPI (blue). Scale bar, 20 µm.

## Data Availability

The data presented in this study are available on request from the corresponding author.

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
