# Peer review of "Red-Emitting Latex Nanoparticles by Stepwise Entrapment of β-Diketonate Europium Complexes"

_ijms, 2022, doi:10.3390/ijms232415954_

Round 1

Reviewer 1 Report

This work synthesized highly monodispersed poly(St-co-MAA) nanoparticles that contain β-71 diketonate Eu3+ complexes by the step-wise process. These nanoparticles can substitute red luminescent organic dyes for intracellular trafficking and cellular imaging agents. I find this manuscript important for future nano-medicine research. The topic falls within the scope of the IJMS journal. That being said, my enthusiasm for endorsing the publication of the manuscript in its present form is tempered by several issues and concerns that must be adequately addressed by the authors as detailed below.

1.    On page 3, Figure 1(c) was not discussed in the main text. Please, provide an explanation and discussion of this figure.

2.    I recommend plotting the particle size distribution for Figure 2 (a-d) and including the average size with standard error/standard deviation. It is hard to conclude if particles are monodispersed from just images.

3.    The XPS data in Figure 5 looks very noisy. It should be baseline corrected and “peak smoothing” should be performed.

4.  There are some mistakes and typos in the manuscript, and English needs to be further improved.

Author Response

Point 1:    On page 3, Figure 1(c) was not discussed in the main text. Please, provide an explanation and discussion of this figure.

Response 1: Thanks for the comments. Following sentence was added and highlighted with red color in main text. “Figure 1 (c) shows photograph images of poly(St-co-MAA) (left) and Eu(TFTB)2(MAA)P(Oct)3@poly(St-co-MAA) (right)  dispersed in water were taken under i) day light, ii) day and UV light (λex at 365 nm) and iii) UV light.”

Point 2:    I recommend plotting the particle size distribution for Figure 2 (a-d) and including the average size with standard error/standard deviation. It is hard to conclude if particles are monodispersed from just images.

Response 2: Thanks for the comments. Based on referee’s comments, the particle size distributions of poly(St-co-MAA) and Eu(TFTB)2(MAA)P(Oct)3@poly(St-co-MAA) nanoparticles were inserted in Figure 2 (e).

Point 3:    The XPS data in Figure 5 looks very noisy. It should be baseline corrected and “peak smoothing” should be performed.

Response 3: Thanks for the comments. Based on referee’s comments, noise in the XPS data has been smoothed and Figure 5 was replaced with a new one.  

Point 4:    There are some mistakes and typos in the manuscript, and English needs to be further improved.

Response 4: Thanks for the comments. Based on referee’s comments, mistakes and typos were corrected.

Reviewer 2 Report

Review Report

I would like to thank all authors of the manuscript for their good and novelty manuscript titled as (Red-emitting latex nanoparticles by stepwise entrapment of β-diketonate europium complexes) which submitted to journal IJMS (ISSN 1422-0067).

1-The manuscript is original and novel as it aims to synthesize the Eu(TFTB)2(MAA)P(Oct)3@poly(St-co-MAA) composed of TFTB, MAA, P(Oct)3 and Eu(III) and to study on the cytotoxic effects of poly(St-co-MAA) and Eu(TFTB)2(MAA)P(Oct)3@poly(St-co-MAA), human hepatic cell (HepG2) line

2-The Presentation of the manuscript is good which attract the Interest to the readers.

3-Minor revision is needed to English language and style.

4-The introduction provide sufficient background and include all relevant references and styled according to the style of the journal.

5- All the cited references relevant to the research.

6- All the cited references relevant to the research.

7- The methods adequately described.

8- The results clearly presented.

9- The conclusions supported by the results.

So, I recommend accepting after minor revision (corrections to minor methodological errors and text editing.

Corrections are:

In Abstract

Corrections are highlighted in the page 1 lines (26,27).

 In  Introduction

Corrections are highlighted in the pages1,2, (37,46,52,58)

In Results and Discussion

Corrections are highlighted in the page 2,3,4,6,7 ,8 ,9,10 lines (107,138,139,140,175,180,183,191,197,234,276,301)

In Materials and Methods

Corrections are highlighted in the page 11,12 lines(350,365,403,414,422)

Author Response

Point 1:    

Corrections are highlighted in the page 1 lines (26,27).

Corrections are highlighted in the pages1,2, (37,46,52,58)

Corrections are highlighted in the page 2,3,4,6,7 ,8 ,9,10 lines (107,138,139,140,175,180,183,191,197,234,276,301)

Response 1: Thanks for the comments. Based on referee’s comments, mistakes and typos were corrected. The corrected sentence was highlighted with red color in main text.